# Distinguishing moral hazard from access for high-cost healthcare under insurance

**Christopher T. Robertson** [1] *, **Andy Yuan** [2], **Wendan Zhang** [2], **Keith Joiner** [2]

**1** James E. Rogers College of Law, University of Arizona, Tucson, Arizona, **2** Department of Economics, University of Arizona, Tucson, Arizona

* robertson@arizona.edu, christophertrobertson@gmail.com

**Data Availability Statement:** All data is available at https://figshare.com/articles/fulldata_stata_dta/9923132.

**Funding:** The authors gratefully acknowledge the funding support from the Center for Management

## Abstract

### Context

Health policy has long been preoccupied with the problem that health insurance stimulates spending ("moral hazard"). However, much health spending is costly healthcare that uninsured individuals could not otherwise access. Field studies comparing those with more or less insurance cannot disaggregate moral hazard versus access. Moreover, studies of patients consuming routine low-dollar healthcare are not informative for the high-dollar healthcare that drives most of aggregate healthcare spending in the United States.

### Methods

We test indemnities as an alternative theory-driven counterfactual. Such conditional cash transfers would maintain an opportunity cost for patients, unlike standard insurance, but also guarantee access to the care. Since indemnities do not exist in U.S. healthcare, we fielded two blinded vignette-based survey experiments with 3,000 respondents, randomized to eight clinical vignettes and three insurance types. Our replication uses a population that is weighted to national demographics on three dimensions.

### Findings

Most or all of the spending due to insurance would occur even under an indemnity. The waste attributable to moral hazard is undetectable.

### Conclusions

For high-cost care, policymakers should be more concerned about the foregone efficient spending for those lacking full insurance, rather than the wasteful spending that occurs with full insurance.

Innovations in Healthcare at the Eller College of Management, from the Rogers College of Law, and from the College of Medicine, all at the University of Arizona.

**Competing interests:** The authors have declared that no competing interests exist.

## Introduction

Although insurance enhances welfare by laying off risk [1] some have estimated that insurance may do more harm than good. [2,3,4,5] Standard health economics theory stipulates that insurance stimulates wasteful spending by insulating patients from the marginal cost of the healthcare they consume. This wasteful spending may take the form of consumption of low-value healthcare, or the failure to pursue lower-cost providers of healthcare, or perhaps even the pursuit of riskier lifestyles that create more need for healthcare in the first place (not the focus of the present paper). On this theory of "moral hazard" insurance creates a collective action problem: individual decisions drive up insurance premiums by consuming care that people would not rationally want if they bore its marginal costs. [6,7] Indeed, a landmark field experiment showed that patients receiving more fulsome insurance spent more on healthcare, with no detectable health b50enefits for median patients. [8] This theory and data have influenced a generation of scholarship in favor of substantial cost exposures in health insurance. For example, leading economists have proposed that patients pay a 50% coinsurance rate with no cap on out-of-pocket exposure. [9]

This work has been deeply influential on U.S. policy, which has eschewed universal and fulsome coverage. [10] When President Bill Clinton proposed an expansion of health insurance, he worried that, "many of us who have had fully paid health care plans have used the system whether we needed it or not without thinking what the costs were." [11] In a contemporaneous survey, economists resoundingly endorsed the moral hazard theory, and many opposed the creation of national health insurance coverage. [12] When President George W. Bush expanded health savings accounts (rather than insurance), he complained that insured patients "really don't know the true costs of medical services they receive." [11] Even President Obama's effort to expand health insurance left families exposed to over $16,300 per year in out-of-pocket expenses (for 2020), which can bankrupt even middle-class families. [13,14]

More recent economic work suggests a more nuanced account of insurance. In a range of clinical situations, Baicker, Mullainnathan, and Schwartzstein note that patients seem to forgo highly efficient healthcare. [15] Even when healthcare is deeply subsidized or free, some diabetics do not adhere to their insulin regimens and some patients with heart disease do not take their beta blockers, for examples. The authors suggest that in these special situations where individuals may be biased against consumption, optimal copays should sometimes be zero, or perhaps negative. They coin a term, 'negative behavioral hazard,' to describe the tendency of individuals to under-consume valuable health care interventions, in contrast to 'positive behavioral hazard,' which is the tendency of individuals to over-consume interventions lacking value.

In this same vein, Brot-Goldberg and associates recently studied a change to greater cost-sharing exposure among a group of highly-paid employees, and found that, while it reduced spending, the intervention did not cause greater price-shopping. [16] In other work, high-deductible healthcare plans have been shown to reduce health spending, but also reduce appropriate preventative care and medication adherence. [17] Recent research suggests that providers may drive spending choices to a much greater degree than patients. [18,19]

Our study adds to the literature in the following two ways. First, we use indemnities as an alternative theory-driven counterfactual in two vignette-based survey experiments with a high-dollar healthcare setting. Second, our analysis of the experimental outcome provides evidence that most of the healthcare spending from insured consumers would occur even under an indemnity. There is no detectable waste attributable to moral hazard triggered by insurance in the high-dollar healthcare context.

## High-cost care and the indemnity counterfactual

To understand the effects of any intervention, such as health insurance, the analyst must compare it to the alternative, or counterfactual. Economists have traditionally focused on uninsurance as the counterfactual for health insurance. Summarizing this "general theory that health economists apply to insurance," leading economist and governmental advisor Mark Pauly explains that "'[m]oral hazard' . . . is generally. . . inefficient, because some of the use of medical care that insurance stimulates (*compared with having no insurance*, when a person pays the full market price) must by definition be care that is worth less to the person than its market price." [20]

While the uninsurance counterfactual is arguably appropriate for low-price consumption (e.g., a generic drug) that is available to both insured and uninsured people alike, it may not be appropriate for high-cost care or poor consumers. In 2015, half of all health expenditures fell on just 5% of the population, and that group spent $50,572 on average, which is also roughly the income for an entire year for a median Americans. [21] In such cases where the care is unaffordable, an uninsured or underinsured patient's "choice" not to consume arguably does not reveal anything about their valuation of the non-consumed goods and services. [22]

John Nyman argues that people buy health insurance not just to spread the risk of large expenses; they seek access to healthcare they otherwise could not afford to consume. [23,24] In addition to insufficient wealth, illness often causes income loss, which thus reduces how much healthcare sick persons could buy out of pocket. [25,26] Nyman's theory has been largely ignored by health economists, [27] perhaps because the theory has not seemed conducive to empirical testing and because it makes it more difficult to make policy-relevant inferences from the fact that health insurance stimulates health spending. The additional spending could be beneficial to welfare, since it provides individuals with the necessary purchasing power to access valuable care.

Yet, increased spending due to insurance may still reflect welfare losses, as consumers do not bear marginal costs. They could, for example, spend $100,000 of the insurer's money to access care worth only $5,000. To distentangle access from moral hazard–a necessity to set optimal insurance policy–a more appropriate counterfactual for insurance is necessary.

In the contemporary U.S. since the 1950s, health insurance benefits are largely paid in-kind to reimburse healthcare providers for the goods and services they provide, rather than paid as cash to the beneficiaries. [28] In contrast, indemnities are a form of insurance, which pays benefits in cash-equivalents to the beneficiary, rather than paying them in-kind to healthcare providers. Scholars have proposed such an indemnity system for healthcare. [29,30,31] Cutler and Zeckhauser have argued that indemnities would be "optimal . . . and the simplest health insurance policy." [32]

Indeed, U.S. automobile insurers sometimes pay cash benefits. [33] Worldwide, there are established markets for indemnities keyed to medical occurrences [34,35] often sold as riders to life insurance policies. [36] These are sometimes known as "dreaded disease" plans, and are largely supplementary to traditional health insurance coverage of medical expenses. Similarly, in the U.S., there is a growing market for "critical illness" insurance plans, which have "are often pitched as an insurance policy for your health insurance policy." [37]

Aside from laying off risk and providing access to otherwise unaffordable care, insurance has other functions, such as bundling insureds into groups, which have greater market power to negotiate prices (with the threat of excluding a provider from a network), compared to individuals who nominally face undiscounted "list" prices. [38] On the other hand, uninsured individuals often receive discounted prices based on lower administrative costs and their ability to pay, which can actually then be less than prices offered to insurers. [39] Although it would be

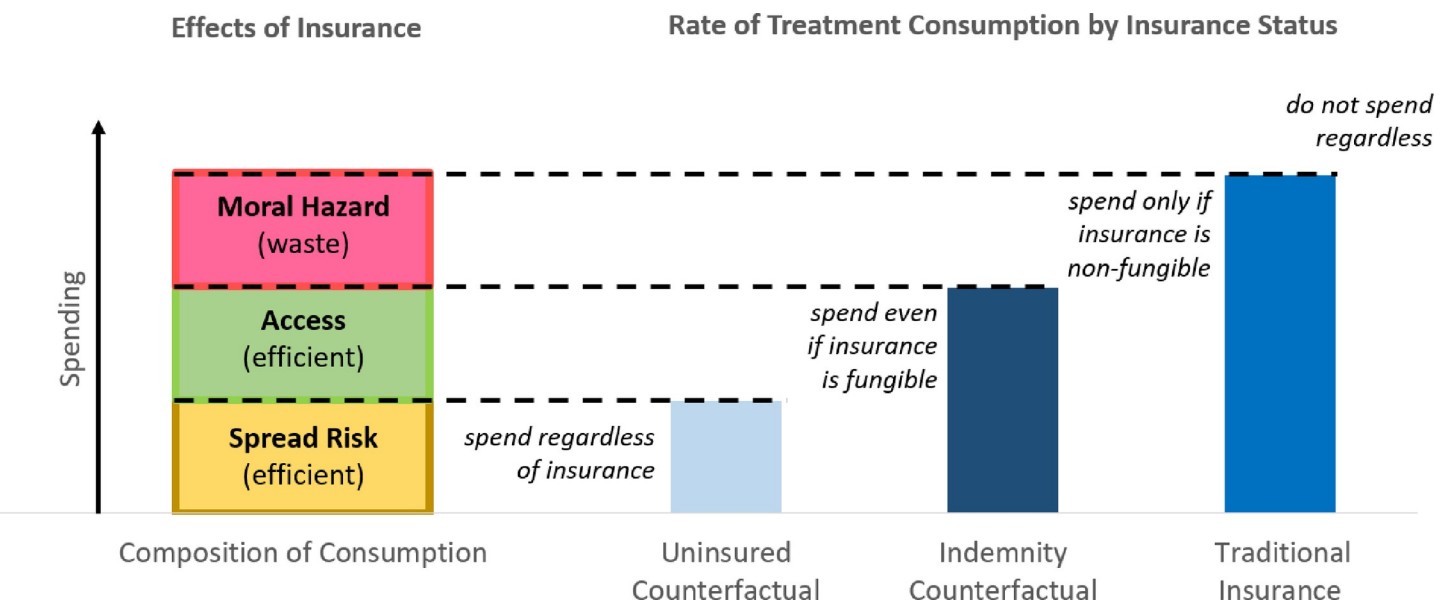

**Fig 1. Theoretical model for decomposing effects of health insurance from two counterfactual conditions.** Values are hypothesized for illustration. Reproduced with permission [40].

more complicated, an insurer could provide this provider-bargaining function, like a discount card, even if it paid benefits to insureds under an indemnity. (In our experimental framework below, we hold constant the price between indemnities, traditional insurance, and uninsured conditions.)

Although they are not used for primary health insurance coverage in the United States today, indemnities are interesting as a potential real-world mechanism for financing health-care and protecting patients from risk. [40] But even if indemnities were completely impractical (perhaps because of fraud or other problems), they are theoretically interesting as a counterfactual for decomposing the effects of health insurance, as shown in Fig 1. Indemnity maintains an opportunity cost for patients, unlike standard insurance. Indemnity also guarantees access to care, unlike uninsurance. The consumption rate under an indemnity thus explains away some of the marginal consumption under traditional insurance, since these individuals would consume even if the insurance were fungible (creating opportunity costs), but do not consume when uninsured (lacking access).

Thus, the analyst needs estimates of consumption for identically-situated patients under uninsurance, indemnity, and traditional insurance. Those with insurance who would consume even if they did not have insurance (as shown in the uninsured counterfactual), are merely using insurance to spread the risks of being exposed to costly care, as in classic economic theory. To estimate the access function of insurance, the analyst measures the difference in proportions on intent to consume between those who are uninsured versus those insured by indemnity, holding all else constant. To measure truly wasteful marginal spending under traditional in-kind insurance, the analyst looks to the difference in rates of treatment consumption between those with indemnities versus traditional insurance. This measure allows an estimate of the moral hazard effect of insurance, since access is controlled in both conditions, but the consumer's price signal is eliminated in traditional insurance.

Nyman and colleagues have recently empirically estimated the size of the access effects of insurance, by examining other income shocks and their effects on healthcare consumption, as an analogy to true indemnities keyed to illness. [41] This population-survey approach is not

keyed to the sorts of expensive healthcare consumption decisions that drive most health spending. In contrast, a simulated indemnity insurance policy would provide a more direct conceptual test. We are aware of no direct empirical studies of the effects of indemnities on medical care consumption, which allow such direct comparisons of the appropriate counterfactuals.

## Materials and methods

We seek to simulate identically-situated patients with no insurance, traditional insurance, or indemnity insurance, in contexts of high-value and low-value healthcare. Lacking archival data on indemnities for primary health coverage and unable to randomize patients to true uninsurance in the field, we constructed a randomized vignette-based survey experiment. Vignette-based research has become a staple of social science generally and useful for simulating discrete choices in healthcare research in particular. [14,42–44] Online populations are an increasingly powerful way to collect large amounts of data from attentive respondents, with high levels of attention and demographic representativeness. [45–47]

### Vignettes

In two experiments, we sought to simulate realistic clinical situations culminating with patients deciding, "...would you take [the treatment]?" We asked respondents to explain that decision in writing as well.

Vignettes were constructed around common medical conditions in the US, for which some costly intervention is presented to and being considered by the patient. The intervention can be carried out on an elective or semi-elective basis: patient decisions are made in a non-emergent setting. For example, Experiment 1 focused on a single clinical situation, with vignettes depicting a hypothetical drug ("Bucarin") as a chemotherapy for the respondent's colon cancer, costing $80,000, on top of the standard regimen the patient will otherwise receive.

Experiment 2 replicated the Bucarin case, and also introduced seven other scenarios as shown in Table 1 (one per respondent). This heterogeneity in case presentations is important because the fraction of moral hazard may depend on whether the treatment directly influences survival from a potentially fatal disease, or whether the treatment choices impacts mobility/ lifestyle, or whether the treatment merely affects personal appearance (to describe three of our cases). Thus across a range of conditions, we tested cases involving a drug for lung cancer, a stent device used for stable coronary artery disease, surgery for gastrointestinal disease, a drug for macular degeneration, a biologic for psoriasis, a knee replacement surgery for degenerative arthritis, and spinal surgery for lower back pain. The vignettes cover a wide range of commonly encountered and expensive medical situations, varying in severity, rate of progression, extent of disability, treatment options, physician preferences, and costs.

The vignettes (available from the authors) were fairly extensive and detailed, ranging from 409 to 576 words (494 on average), with an additional 551 words presented for informed consent, writing prompts, etc. All vignettes contained the following components, corresponding to the information used in clinical decision making by patients and providers: history of present illness including medical management of the condition to date, past medical history relevant to the condition, diagnostic testing data where appropriate, and costs of the procedure. All patients are seen by specialists for the respective condition. These specialists provide informed consent, covering potential benefits, side effects and risks of the intervention. Total costs for interventions are typical for such interventions in the US, ranging from $15,000 to $125,000, as shown in Table 1.

**Table 1. Summary of experimental vignettes.**

| Disease or Condition | Proposed Treatment / Baseline | Proposed Treatment | |
|---|---|---|---|
| | | High Value Manipulation | Low Value Manipulation |
| **Cancer: Adenocarcinoma of the colon** | $80k novel drug / standard therapy | Drug has been approved by the FDA for colon cancer and studies show it improves chances of survival | Drug has not been approved for use in colon cancers. Oncologist has had good experiences using the drug off-label |
| **Cancer: Non-small-cell lung cancer** | $125k novel drug / home hospice | Drug increased average survival by 8 months | Drug stopped tumors growth for 4 months but did not show any survival benefit |
| **Cardiovascular: Coronary Artery Disease** | $55k drug-eluting stent / medical therapy | The FDA has approved the stent to prevent heart attacks because it improves survival | Stent is not approved for prophylactic use (prior to heart attack), but is recommended off-label, with no survival benefit |
| **Gastrointestinal Disease: GERD** | $45k gastric reflux surgery / medical and lifestyle therapy | Three-quarters of patients experience relief of their symptoms | Half of patients experience relief of symptoms, but side effects can be substantial and patient is not optimal surgical candidate |
| **Age-Related Macular Degeneration (AMD)** | $15k novel drug / established drug | Novel drug improves vision in 2/3 of cases compared to 1/3 of cases with established drug | There is less data on the very new novel drug and doctor recommends sticking with the established drug |
| **Skin/Autoimmune disease: Psoriasis** | $20k biologic drug / standard medical therapy | Two-thirds of patients who take the drug report clear skin, and low dose minimizes potential severe risks | One-third of patients who take drug report clear skin; side effects can be severe, including cancer |
| **Orthopedic Disease: Degenerative Arthritis** | $70k total knee replacement surgery / medical management | Orthopedic surgeon finds instability in right knee which requires surgery, which should occur now | Orthopedic surgeon finds no instability but predicts surgery will eventually be needed; but arthritis is still mild to moderate |
| **Neurologic Disease: Degenerative Lower Spine Disease** | $85k spinal surgery / medical management | New surgery removes pain and disability in two-thirds of patients and most patients struggle to comply with alternative medical treatments | New surgery removes pain and disability in one-third of patients, which is equivalent to the results of intense medical treatment including physical therapy |

Every respondent was offered the stipulated baseline treatment, which they would receive if they declined the proposed treatment. Respondents were randomly assigned to consider one high value or low value treatment against that baseline.

## Manipulations

We varied both the type of insurance and the value of care (high or low, with each subject receiving a recommendation for one case). High and low value scenarios are broadly consistent with the range of outcomes described in the medical literature. High value scenarios reflect the more certain and/or more effective outcomes, while low value scenarios reflect the less certain and/or less effective outcomes. As an example of a value manipulation, in Experiment 1, the high-value condition is for an FDA-approved drug, with proof of safety and efficacy for the patient's condition, while the low-value condition offers an off-label use for a condition not specifically considered by the FDA, with no proof of safety and efficacy for the patient's specific condition. In the low-value condition, the drug is nonetheless recommended by the oncologist, as is quite common in the real world [48].

In manipulating the respondent's insurance status, we asked them to imagine that they are either (a) uninsured, (b) with traditional insurance, or (c) insured in a cash indemnity program. For the uninsured conditions, the text provided:

"Please imagine that **you currently do not have health insurance**, however you do have the same amount of wealth and credit that you actually have in the real world (the amount you provided in your answer above). So, only if you have enough money available, you can pay for the Bucarin yourself. Think back to your answer to the purchasing power above. If you do have that much purchasing power, please consider however, other things you may prefer to do with the money. Please be realistic."

For the traditional insurance conditions, the text provided:

"Please imagine **you have full health insurance**, and you have already met your annual deductible, so there is no additional cost to you out of pocket, if you choose to take [the treatment]. In deciding, please be realistic."

The indemnity was described as:

Please imagine you have full health insurance, and you have already met your annual deductible. Your insurance company has a unique program, in which they cover special drugs with a cash payment, called an indemnity (similar to car insurance). Because your physician has indicated that you are a candidate for [the treatment] **you receive a $125,000 cash payment from your insurance company. You receive the money regardless of whether you decide to take** [the treatment]. You decide whether you spend the cash on the [the treatment] or instead keep the money for whatever other purposes you may choose. In deciding, please be realistic.

Notably, the indeminity script uses the word "cash" three times. In the vignettes, the insurance manipulations were highlighted in boldface text, as shown here. In both of the insurance conditions, there is no cost-sharing required, on the theory that the patient has already reached her annual maximum through the cascade of healthcare that culminates in this pivotal decision, a not-uncommon phenomenon. [22]

Based on the design described above, Experiment 1 had six conditions (3 insurance types x 2 treatment values), in between-subjects design with equal allocation across all variables and levels thereof. Experiment 2 yielded 48 conditions (3 insurance types x 2 treatment values x 8 clinical scenarios), in between-subjects design with equal allocation across all variables and levels thereof. Manipulations were fully crossed in factorial design to maximize statistical power. [49,50]

## Respondent engagement

On the informed consent page, we warned all respondents that they may be quizzed on the material, and that they should be careful. To increase engagement and internal validity, we also included several mandatory writing prompts that interrupted the vignette. Halfway through the scenario, we asked, "Please refer to the story specifically, and share a sentence describing how you would feel in this medical situation, and your thoughts about your treatment options described so far." At the end of the scenario in Experiment 2, we also asked, "On the next page you will share your decision. First, now please write two sentences putting in your own words, (1) your understanding of the treatment options, and (2) your insurance situation, as described above."

A research assistant (RA) manually reviewed whether respondents provided minimally meaningful responses to the writing prompts. The RA was blinded to respondents' experimental conditions. Those providing junk data were removed.

To further ensure engagement, in both experiments, after the survey on a separate page, we asked all respondents to identify their insurance type in the vignette. Those answering this manipulation-check incorrectly were screened. In Experiment 2 we similarly required a correct answer to a disease-type query, and screened those failing.

## Research populations and data quality

In Experiment 1, we recruited human subjects from a platform (Mechanical Turk ("Mturk")) provided by Amazon.com to collect 613 valid observations. To ensure engagement in the

materials and reduce the risk of the work being completed by non-human "bots", we pre-qualified respondents who were successful in at least 90% of prior tasks they had attempted. [47] We also used TurkPrime to block duplicate IP addresses and verify worker country location (United States), and we paid $1.25 to each worker, who (on the median) spent 8.0 minutes to complete the task. The research population was reasonably diverse and randomization successfully distributed observable covariates across experimental conditions (S1 Table). After applying our manipulation checks, RA review of the qualitative responses found no junk data for exclusion.

In Experiment 2 we expand generalizability by including a range of seven other clinical scenarios; quadruple the sample size; replicate with a different respondent population (Survey Sampling International (SSI)); and recruit a demographically representative pool, tracking U. S. Census distributions of age, gender, and household income, with successful randomization across conditions (S2 Table). SSI uses multiple recruitment and incentivization methods, described in the Supplemental materials. In Experiment 2, we began with 2400 valid observations, defined as those who passed the insurance-type and disease-type manipulations. A blinded RA reviewed qualitative responses, and we eliminated 44 respondents who did not engage meaningfully in the task, yielding 2,356 responses for analysis.

With the above pre-qualification screening and a blinded manual review of qualitative responses, fraudulent responses, including bots are unlikely to be contaminating the final data in our analysis. Due to the randomized controlled trial design, any such noise would bias the analysis towards finding null result. This turns out to be not the case as we see a sharp contrast among responses under two of three different insurance types and both levels of treatment value. Additionally, the estimations are consistant across two experiments, conducted with different populations.

## Covariates and analysis

We collected standard demographics and respondents' actual insurance status (as distinct from the manipulated insurance type descried above) for use as controls. Prior to the vignettes we also asked each person for their maximum purchasing power (i.e., cash, formal and informal credit, sellable assets) that could be used "for a very important purchase . . . if absolutely necessary to make such a purchase, if you absolutely had to do so." To ensure some level of introspection on this point, we asked respondents qualitatively "explain briefly how you would come up with that amount of money. From what sources would you draw to reach that amount? What might you have to give up?" In some of our models, we used the quantiative amount as an "impossibility screen" to evaluate subsequent respondent choices to consume healthcare in the uninsured condition. The vignettes do not ask respondents to imagine different purchasing powers than they have in the real world; we exploit this natural variation which may be salient to respondents. (As it happens, this screen does not substantially change our results.)

In our regression models, we created subsets of the data to test the theory-driven contrasts (uninsurance versus indemnity to test the access function of insurance, and indemnity versus traditional insurance to test for moral hazard). To produce regression-adjusted percent of patients consuming the healthcare across condition (Fig 2), the main specification employs a linear probability model, for ease of interpreting coefficients (S3 & S4 Tables). The Supplemental Information also presents logit and probit models, which generally lead to similar results. For robustness, we there present analyses with and without controls, and with and without data screens.

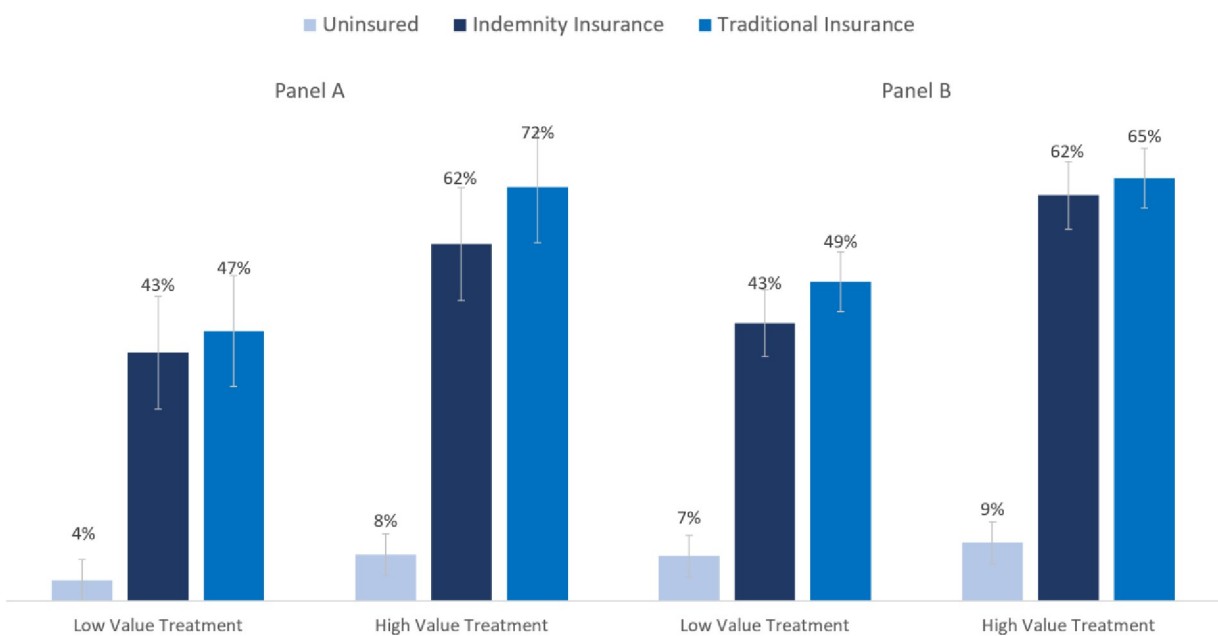

**Fig 2. Percent consuming treatment by insurance type and value of treatment with bootstrapped 95% confidence intervals across two study populations (Panels A & B).** Panel A is N = 613 respondents provided by Amazon.com, adjusted for demographics, in a cancer vignette, either on-label (high value) or off-label (low value). Panel B is N = 2,356 respondents from Survey Sampling International, representative to U.S. Census, by age, gender, and income; randomized across eight vignettes and adjusted for demographics and vignette. Both samples exclude those failing insurance-type manipulation check and purchasing-power impossibility check. As shown in Table 2, using a Mann-Whitney U Test and T-test, the differences between uninsurance and either type of insurance are highly significant; the differences between insurance types are not significant.

As shown in Fig 3, to identify a likely counterfactual for each individual respondent, we also deployed Bayesian simulation analogue to the Peters-Belson method. [51,52,53] This approach provides greater precision, of particular importance for interpreting potential null effects.

## Results and discussion

Participants' intent to consume the proposed treatment was very responsive to some variations on insurance type and treatment value (Fig 2 and Table 2). We find virtually no spending when respondents lack health insurance, but very substantial spending in the traditional insurance condition. With insurance, respondents were about five times more likely to consume the preferred healthcare. On the other hand, although the sign for moral hazard is in the theorized direction, there were no significant differences in consumption between those in the indemnity and traditional insurance conditions. This null result persisted in the replication, and pooling across seven additional vignettes, and regardless of whether we applied regression controls (Table 3) and manipulation checks and alternative logit and probit models (S3 and S4 Tables). The main results are as follows.

We did not power our study to explore specific hypotheses related to individual clinical conditions; instead we included these manipulations to improve generalizability. Nonetheless, the Supplemental Information displays regression results for individual scenarios (S5–S12 Tables). We found that respondents tended to be sensitive to the value of the healthcare proposed–just as the factor was highly significant in our pooled models, treatment value made a detectable difference (at the 0.05 level) to respondent decisions in 5 out of 9 of the individual disease experiments (including Experiment 1, its replication in Experiment 2, and seven

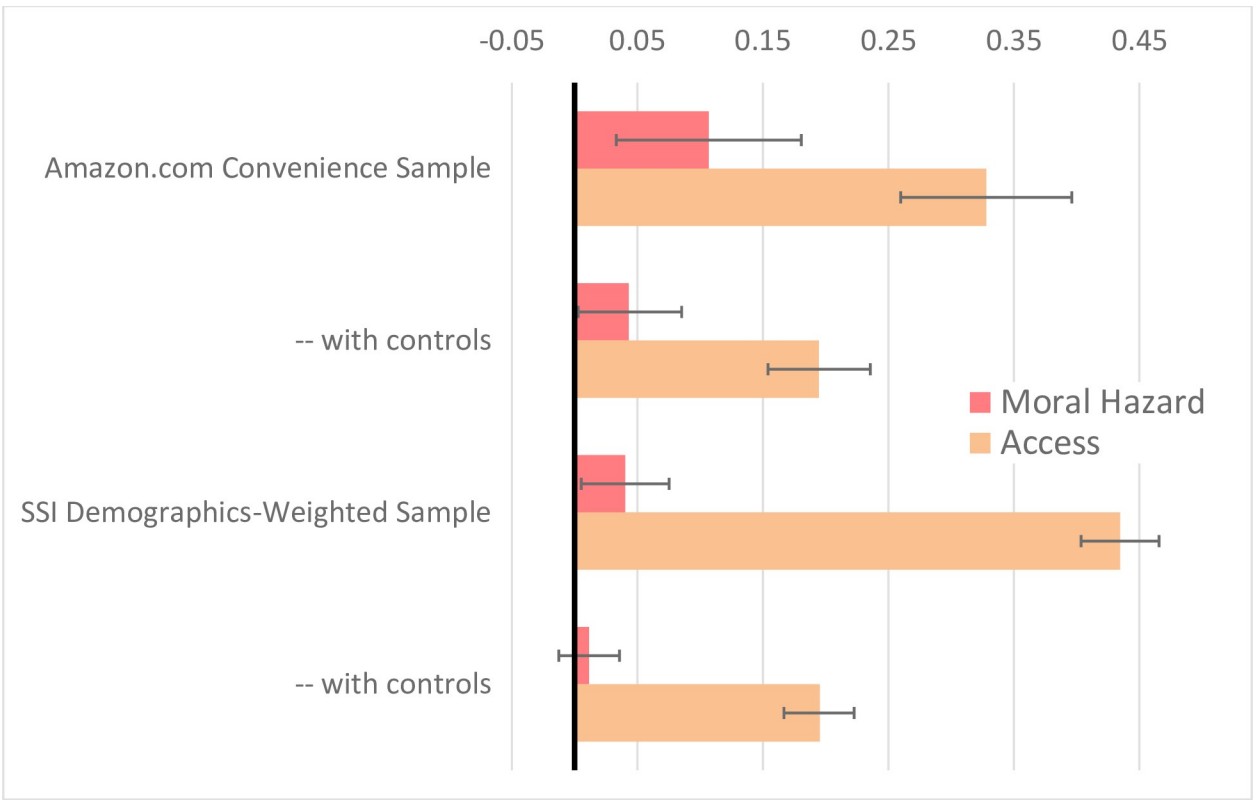

**Fig 3. Marginal effects of insurance (moral hazard and access) on intent to consume healthcare under Bayesian Monte Carlo estimation method, using uninsurance and indemnity counterfactuals (with 95% Confidence Intervals).** Amazon.com convenience sample is N = 613 respondents from Mturk, in a cancer vignette, randomized to either on-label (high value) or off-label (low value) proposed treatment. SSI Demographics-Weighted Sample is from Survey Sampling International, N = 2,356, representative to U.S. Census, by age, gender, and income; randomized across eight vignettes in high or low value. Controls adjust for demographics, value of proposed healthcare, and vignette. Both samples exclude those failing insurance-type manipulation check and purchasing-power impossibility check. When controls are applied, moral hazard is not statistically significant under either sample. Access is significant under both samples.

others). Access was the key function of insurance: in 8 out of 9 of the scenario cases (all but the psoriasis scenario), those with health insurance consumed more, even with fungible insurance that had an opportunity cost. In contrast, we found no moral hazard detectable (at the 0.05 level) in any of our scenarios (0 out of 9) when controls are included. Just as in our highly-powered pooled results, we found no moral hazard that could be distinguished from the null (at the 0.05 level). Considering the interaction effect, we found that the fungibility of insurance in the indemnity also did not make respondents detectably more sensitive to the value of the proposed treatment in any of the scenarios (0 out of 9), the same result as in our overall model.

## Strengths and limitations

Our randomized design blinds respondents and avoids certain forms of response bias, while solving for endogeneity and internal validity, thus allowing causal inferences, unlike typical surveys. Our vignette-based approach allows study of indemnity insurance plans, which do not presently exist as a form of primary health insurance. The realistic vignettes drafted by a physician, represented a wide range of cases of high-cost care, which drive most health spending, but where it is infeasible to field experiment due to the ethical and practical difficultly of deny insurance for such life-saving care. This is precisely the domain where the access function of insurance may be most important [24] but precisely where we cannot extrapolate from

**Table 2.  Bivariate tests of statistical significance for differences in treatment consumption between insurance types, with subsets for treatment value (Low, High, Both).**

**Panel A: Online Convenience Sample Experiment (Amazon.com)**

|  | Uninsured minus Indemnity | | | Indemnity minus Traditional | | |
|---|---|---|---|---|---|---|
| Value of Healthcare | Low | High | Both | Low | High | Both |
| Mann-Whitney U test (CLES[a]) | 0.389*** | 0.303*** | 0.345*** | 0.482 | 0.451 | 0.468 |
| T-test (Mean Difference[b]) | -0.222*** | -0.394*** | -0.309*** | -0.037 | -0.098 | -0.065 |
| Obs N1,N2 | 101,100 | 103,102 | 204,202 | 100,105 | 102,102 | 202,207 |

**Panel B: Online Census-Weighted Experiment (SSI Sample)**

|  | Uninsured minus Indemnity | | | Indemnity minus Traditional | | |
|---|---|---|---|---|---|---|
| Value of Healthcare | Low | High | Both | Low | High | Both |
| Mann-Whitney U test (CLES[a]) | 0.400*** | 0.331*** | 0.366*** | 0.468 | 0.487 | 0.480 |
| T-test (Mean Difference[b]) | -0.201*** | -0.339*** | -0.268*** | -0.064 | -0.027 | -0.040 |
| Obs N1,N2 | 366,351 | 425,356 | 791,707 | 351,452 | 356,406 | 707,858 |

[a] CLES stands for the Common Language Effect Size, which is the probability that variable for the first group is larger than variable for the second group.

[b] Mean Difference for each T-test is the mean of the second group subtracted from the mean of the first group.

"***" significant at 0.1% level

"**" significant at 1% level

"*" significant at 5% level.

studies of routine, inexpensive healthcare. [54] In this domain, discrete choice experiments, like the one we fielded, provide a useful approach. [46,47]

We replicated our findings in two independent and diverse research populations. One provided incentives for respondents to pay close attention, and another provided a more demographically-representative sample.

One interesting feature of our vignette approach could be counted as a strength (for theoretical clarity) and a weakness (for realism): we excluded implicit forms of insurance that exist in the United States. [55] Specifically, for the respondents in the uninsured conditions, we did not make salient the fact that they could get some healthcare in hospital emergency rooms, without showing proof of payment, due to a Federal law called EMTALA. Further, although we did ask them to consider their access to credit when estimating their maximum purchasing power, we did not make especially salient the fact that they could rack up medical bills and ultimately fail to pay them, perhaps even discharging them in bankruptcy. These implicit forms of insurance have many disadvantages and inefficiencies, but they are one reason that prior studies of explicit insurance have not found major benefits for health. Thus, the uptake rates shown in our uninsured conditions are likely unrealistically low, since in the United States access would often in fact be provided through these implicit insurance mechanisms (making nobody truly uninsured in this sense). Nonetheless, as the analytic baseline for standard economic theory [20], the truly uninsured patient is the relevant construct to simulate, since only such a person is fully incentivized to make cost-benefit tradeoffs.

Our methodology has other limitations. Most importantly, respondents may not have fully grasped the importance of the insurance type manipulation. Since indemnities are not commonly used in health insurance, notwithstanding the bold type used in our case presentations, respondents may have failed to appreciate that they would receive a cash payment, which they could spend on anything else they preferred. Nonetheless, respondents noticed and reacted to the relatively subtle and complicated manipulations concerning the value of the proposed treatment, with statistical significance at the .01 level. Moreover, we tested respondents *ex post* to determine whether they remembered which type of insurance plan they were in and we

**Table 3. Linear probability models on intent to consume treatment.**

**Panel A: Online Convenience Sample Experiment (Amazon.com)**

| | Uninsured vs Indemnity Insurance (Access) | | Traditional Insurance vs Indemnity (Moral Hazard) | |
|---|---|---|---|---|
| Indemnity Insurance | 0.222*** | 0.225*** | | |
| | (0.064) | (0.067) | | |
| Traditional Insurance | | | 0.037 | 0.055 |
| | | | (0.068) | (0.070) |
| Value of Healthcare | 0.015 | 0.012 | 0.188** | 0.180* |
| | (0.064) | (0.066) | (0.068) | (0.071) |
| Indemnity X Value | 0.172 | 0.165 | | |
| | (0.090) | (0.094) | | |
| Traditional X Value | | | 0.061 | 0.034 |
| | | | (0.096) | (0.099) |
| Constant | 0.208*** | 0.146 | 0.430*** | 0.248 |
| | (0.045) | (0.202) | (0.049) | (0.223) |
| Controls | NO | YES | NO | YES |
| R-squared | 0.121 | 0.128 | 0.054 | 0.073 |
| N | 406 | 390 | 409 | 396 |

**Panel B: Online Census-Weighted Experiment (SSI Sample)**

| | Uninsured vs Indemnity Insurance (Access) | | Traditional Insurance vs Indemnity Insurance (Moral Hazard) | |
|---|---|---|---|---|
| Indemnity Insurance | 0.201*** | 0.218*** | | |
| | (0.035) | (0.036) | | |
| Traditional Insurance | | | 0.064 | 0.032 |
| | | | (0.035) | (0.036) |
| Value of Healthcare | 0.058 | 0.076* | 0.196*** | 0.194*** |
| | (0.033) | (0.034) | (0.037) | (0.038) |
| Indemnity X Value | 0.138** | 0.121* | | |
| | (0.048) | (0.050) | | |
| Traditional X Value | | | -0.037 | -0.016 |
| | | | (0.050) | (0.051) |
| Constant | 0.227*** | 0.338*** | 0.427*** | 0.676*** |
| | (0.024) | (0.095) | (0.026) | (0.105) |
| Controls | NO | YES | NO | YES |
| R-squared | 0.097 | 0.167 | 0.033 | 0.107 |
| N | 1,498 | 1,377 | 1,565 | 1,427 |

Standard errors shown in parentheses.

"***" significant at 0.1% level

"**" significant at 1% level

"*" significant at 5% level. Controls include demographics and vignette-type.

excluded those who failed. In particular, respondents in the indemnity condition had to correctly respond that they were in the condition where the insurer would pay them "cash", which they could choose to spend on the treatment, or not.

Notwithstanding general findings that vignette research can be valid if done appropriately [14,42–47], a variety of biases may infect vignette decisions compared to real world decisions. For example, in the real world, perhaps respondents would be more likely to feel the pressure of alternative needs to spend money, especially when sick and unable to work. Or respondents may be more likely accept their physicians' recommendations, in the heat of a clinical

encounter. Across our 16 vignettes, with eight diseases and two levels of quality, driving much different base rates of consumption, we see consistent dynamics for our experimental manipulations.

We also simplified the healthcare system in various ways. We did not simulate the various sorts of cost-sharing profiles that could be applied to traditional insurance or indemnities. In fact, large-dollar spending often happens after cost-sharing maxima have been exhausted by prior spending. [22] We also used individuals' real-world incomes and maximum purchasing powers, but randomly assigned them to experimental insurance statuses, without regard for their capacity to afford insurance premiums (a remarkably complicated question, given the explicit and implicit premium subsidies that exist in the employer and individual healthcare markets). The distinction between high value and low value care in some vignettes was more stark than would typically be the case in actual medical practice. Nonetheless, our vignettes provided a clean test of consumption decisions in the idealized-but-complex clinical situations presented.

## Conclusions

Fig 3 summarizes our primary findings. We were consistently unable to find a significant effect of moral hazard waste, and the upper ranges of our confidence intervals help rule out the hypothesis that the problem is substantial in the experimental sample. In contrast, we consistently found a substantial effect of insurance to provide access to expensive healthcare. These suggest that moral hazard in the healthcare sector may not be large, but the benefit of insurance providing access to expensive healthcare could be substantial.

This research project sheds light on moral hazard, which has preoccupied health economics and U.S. health policy for half a century. By testing a novel counterfactual of indemnity insurance, we distinguished the access function of health insurance from waste, and thus informed longstanding debates about how fulsome health insurance coverage should be.

Importantly, we focus on expensive interventions, associated with serious illness, which drives aggregate health spending. [21] Our data suggest that in these circumstances, moral hazard waste is not substantial, since the vast majority of spending stimulated by insurance would happen with fungible insurance, which preserves a price signal for consumers. We make no claim that these findings can or should be extrapolated directly to more routine care decisions.

Healthcare is far from an ideal market: it is rife with wasteful spending and infected by all sorts of market failures, including the misaligned incentives of healthcare providers. Our data show what we know obtains in the real world: healthcare consumers are often willing to consume low-value care, when their physicians recommend it. Yet, our data show that the problem exists regardless of whether patients have a traditional insurance policy that occludes the price of care or an indemnity policy that makes the price salient along with the opportunity costs of consumption. Thus, our data helps to pinpoint the problem, and helps us understand where future policymaking should focus. For example, our study suggests that it will be more effective to align incentives of providers with health and thrift, rather than placing more risk on patients.

Our data is consistent with macro-economic data, showing that the U.S. has some of the highest healthcare spending *along with* very high levels of out of pocket patient cost-exposure. Countries with insubstantial patient cost exposure have some of the most efficient healthcare systems, delivering impressive health outcomes at lower overall costs than in the United States [56]. Our data is also consistent with research findings that providers drive consumptions much more than patients [18,19] and that patients exposed to cost decline both high-value and low-value are alike. [15,16,17] These studies have been unable to distinguish the access function of insurance in particular.

The lack of healthcare consumption associated with uninsurance and underinsurance appear to be a much more substantial problem than moral hazard associated with fulsome insurance. Health policy should accordingly reconsider patient-cost exposure and pursue other measures to reduce the prevalence of low-value healthcare.

## Supporting information

**S1 Table. Demographics split by insurance type manipulation.**
(DOCX)

**S2 Table. Demographics split by insurance type manipulation.**
(DOCX)

**S3 Table. Online convenience sample experiment (amazon.com)–linear probability models on intent to consume treatment.**
(DOCX)

**S4 Table. Online census-weighted experiment (SSI Sample)–regressions on intent to consume treatment.**
(DOCX)

**S5 Table. Cancer: Adenocarcinoma of the colon.**
(DOCX)

**S6 Table. Cancer: Non-small-cell lung cancer.**
(DOCX)

**S7 Table. Cardiovascular: Coronary artery disease.**
(DOCX)

**S8 Table. Gastrointestinal disease: GERD.**
(DOCX)

**S9 Table. Age-Related Macular Degeneration (AMD).**
(DOCX)

**S10 Table. Skin/Autoimmune disease: Psoriasis.**
(DOCX)

**S11 Table. Orthopedic disease: Degenerative arthritis.**
(DOCX)

**S12 Table. Neurologic disease: Degenerative lower spine disease.**
(DOCX)

**S1 Data.**
(DOCX)

**S2 Data.**
(ZIP)

## Acknowledgments

The authors appreciate the research assistance of Ellen Stark. The project benefitted from review of preliminary drafts of the protocol and/or manuscript by John Nyman, Gautam Gowrisankaran, William Sage, Victor Laurion, Jamie Robertson, Jim Greiner, Tara Sklar, Simone

Sepe, Anna Rita Bennato, Govind Persad, and anonymous peer reviewers. The project benefitted from feedback at University of Arizona's QuantLaw and University of Arizona's Interdisciplinary Experimental Readings Group, University of Minnesota's Symposium on Drug Prices, the American Association of Law Schools Insurance Law Section, the Philosophy, Politics, and Economics Program at University of Pennsylvania, and the Conference on Empirical Legal Studies. Fig 1 is also forthcoming as part of a book published by Harvard University Press.

## Author Contributions

**Conceptualization:** Christopher T. Robertson, Keith Joiner.

**Data curation:** Andy Yuan, Wendan Zhang.

**Formal analysis:** Christopher T. Robertson, Andy Yuan, Wendan Zhang.

**Funding acquisition:** Christopher T. Robertson, Keith Joiner.

**Investigation:** Christopher T. Robertson.

**Methodology:** Christopher T. Robertson, Andy Yuan, Keith Joiner.

**Project administration:** Christopher T. Robertson.

**Supervision:** Christopher T. Robertson.

**Visualization:** Christopher T. Robertson, Andy Yuan, Wendan Zhang.

**Writing – original draft:** Christopher T. Robertson, Keith Joiner.

**Writing – review & editing:** Christopher T. Robertson, Andy Yuan, Wendan Zhang, Keith Joiner.

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
