## [Decision Letter · Decision Letter 0]

12 Dec 2019

PONE-D-19-27471

Distinguishing moral hazard from the access function of health insurance: A vignette experiment

PLOS ONE

Dear Dr. Robertson,

Thank you for submitting your manuscript to PLOS ONE. After careful consideration, we feel that it has merit but does not fully meet PLOS ONE’s publication criteria as it currently stands. Therefore, we invite you to submit a revised version of the manuscript that addresses the points raised during the review process.

I am sorry for the long duration of the review process. Obtaining referees for the study, perhaps due to the methodology differences to others researching in similar areas, was more difficult than normal. In the end I had 1 outside reviewer and I reviewed the paper in detail myself as well.  Both the reviewer and I am favorable to your paper.  If you decide to submit a minor revision of the paper please include with your resubmission a detailed response to the concerns and points of the reviewers.  Find here my own comments:

Major:

The paper could be improved by adding some discussion of what you are doing into the introduction. It reads as a lit review or state of healthcare but does not give any indication of what the paper will be about.There are some downsides that need to be addressed in greater detail in the paper. For instance, in the introduction the paper neglects to discuss the idea of negotiated prices. Traditionally insured individuals and uninsured individuals do not only differ in “access” and “moral hazard” components of spending (as in Fig 1) but also differ in what the expected cost of a given healthcare good costs. Insured individuals gain not only the funding for a procedure but also gain the pooled bargaining power of the insurance company. Uninsured individuals in some (most?) cases would face not only the out-of-pocket costs that the insurance company would alternatively pay for but also a second premium for not having bargaining power. The same procedure would likely cost them more to purchase.  How would/do indemnities work on this margin? Do the companies that back the indemnities negotiate healthcare prices as traditional insurers do? Citations here are needed as well.Please provide your vignettes in an appendix.P11: I like that you provide in-text the wording for the indemnity program. I would also like to see how the uninsured and traditional insurance differs in-text or as a footnote.P12: Clarify for experiment 2 whether this is a between-subject or within-subject design.  If a between subject design, please indicate how the 2400 observations are divided between the 3x2x7 treatments.  Likewise, for Experiment 1 how were the 613 observations divided between thr 3x2 treatments?P12: can you also tell us how participants were paid in MTurk & your other sample?  Flat rate? How much?  How did you address the concern of bots on MTurk?Figure 2 Note: In the last sentence where you indicate the significant differences or lack of significant differences, please provide the test you are using.Please move the table of your main regression analysis to the main text.I would like to see some non-parametric analysis of raw data as seen in figure 2.  Mann-Whitney is probably ideal.  You are making claims of differences and no differences but are not utilizing the most direct method of analyzing your experimental data and instead base everything off of regressions.You attach your stata dta file but not your do file to run your analysis. Please include that in your revision.

Minor:

Need citation and numbers for claim that high-$ healthcare drives most aggregate healthcare spending in the USSome typos remaining in paper (for instance, no brackets around citation #10, #30, or #46)You refer to the difference in indemnity and traditional insurance as moral hazard throughout paper but in your tables you call it “waste”. This is somewhat of a disconnect.  Why not just call it “M.Hzd” in the parentheses “Insurance (M.Hzd)” to make things more clear?P 15: you say “In contrast, we found no moral hazard detectable (at the 0.05 level) in any ofour scenarios (0 out of 9). ” this is not exactly true. In table S12 b you have a 5% level significance when controls are excluded. If you want to make this claim in the paper you need to add the qualifier that you are meaning “when controls are included”.P19 “rule out the hypothesis that the problem is substantial.” You may want to add a qualifier that this is “in the experimental sample”. You cannot “rule out” that there is a moral hazard in the healthcare sector at large, you can only say that your data suggests that it is not large.

We would appreciate receiving your revised manuscript by Jan 26 2020 11:59PM. To enhance the reproducibility of your results, we recommend that if applicable you deposit your laboratory protocols in protocols.io, where a protocol can be assigned its own identifier (DOI) such that it can be cited independently in the future. For instructions see: http://journals.plos.org/plosone/s/submission-guidelines#loc-laboratory-protocols

We look forward to receiving your revised manuscript.

Kind regards,

Jason Anthony Aimone

Academic Editor

PLOS ONE

2. We note that Figure(s) 1 in your submission contain copyrighted images. All PLOS content is published under the Creative Commons Attribution License (CC BY 4.0), which means that the manuscript, images, and Supporting Information files will be freely available online, and any third party is permitted to access, download, copy, distribute, and use these materials in any way, even commercially, with proper attribution. For more information, see our copyright guidelines: http://journals.plos.org/plosone/s/licenses-and-copyright.

a)    You may seek permission from the original copyright holder of Figure(s) [#] to publish the content specifically under the CC BY 4.0 license.

3. Please ensure that you include a title page within your main document. You should list all authors and all affiliations as per our author instructions and clearly indicate the corresponding author.

Reviewers' comments:

Reviewer's Responses to Questions

**Comments to the Author**

1. Is the manuscript technically sound, and do the data support the conclusions?

Reviewer #1: Yes

2. Has the statistical analysis been performed appropriately and rigorously? 

Reviewer #1: Yes

3. Have the authors made all data underlying the findings in their manuscript fully available?

Reviewer #1: Yes

4. Is the manuscript presented in an intelligible fashion and written in standard English?

Reviewer #1: Yes

5. Review Comments to the Author

Reviewer #1: This is an innovative paper addressing an important question. The authors present very well the background underpinning their research which helps motivate a very critical question in the literature. They apply a proven-sound methodology. The authors explain their results eloquently and in manner that is accessible to a wide reading audience. The policy implications of this paper can be extremely important and/or start a fruitful string of research in this area.

6. PLOS authors have the option to publish the peer review history of their article (what does this mean?). If published, this will include your full peer review and any attached files.

Reviewer #1: No

---

## [Editor Report · Decision Letter 1]

10 Mar 2020

PONE-D-19-27471R1

Distinguishing moral hazard from access for high-cost healthcare under insurance

PLOS ONE

Dear Dr. Robertson,

Thank you for submitting your manuscript to PLOS ONE. After careful consideration, we feel that it has merit but does not fully meet PLOS ONE’s publication criteria as it currently stands. Therefore, we invite you to submit a revised version of the manuscript that addresses the points raised during the review process.

I am ready to accept this paper but there are just a few small edits, that are very minor, but I  need to mark the paper as minor revision in order to provide an easy way for you to upload those edits.  The direction of these things is not relevant to the scientific rigor of the study but the clarity is needed for the paper.

First, Figure 2 and Table 2 are a bit confusing. It appears Indemnity has the highest uptake (consumption) of all three categories in each situation.  This would imply a negative moral hazard rather than a positive moral hazard directionally (even if not significant).  The labels of the bars and orders of the bars seem to have changed since the first version of the figure, maybe something got mixed up?  Table 2 then is hard to interpret the signs of the values in the table (first group minus second group)? Clarifying better what the values in the table are and what is being subtracted from what would be very helpful.    

Second, Table 3 is a bit confusing. The variable "insurance" is self explanatory for the uninsured vs indemnity regression but it is not self explanatory for the indemnity vs the traditional insurance regression. For all of these regression tables, If the variable is indemnity, use that label rather than insurance. If it is "Traditional Insurance" please use that label (and make a separate row to distinguish from indemnity in the other regression model).  Added to the confusion from Figure 2/Table 2 these regressions become hard to understand.

Finally, you say in your response document “Such details about payment and recruitment are not available from the SSI pool; those functions are outsourced to the provider.”  This is not necessarily a problem but should just be mentioned. Readers need to know whether they were incentivized in their responses or did it without compensation. I’d recommend contacting your provider and asking them about the incentive

We would appreciate receiving your revised manuscript by Apr 24 2020 11:59PM. To enhance the reproducibility of your results, we recommend that if applicable you deposit your laboratory protocols in protocols.io, where a protocol can be assigned its own identifier (DOI) such that it can be cited independently in the future. For instructions see: http://journals.plos.org/plosone/s/submission-guidelines#loc-laboratory-protocols

We look forward to receiving your revised manuscript.

Kind regards,

Jason Anthony Aimone

Academic Editor

PLOS ONE

---

## [Author Response · Author response to Decision Letter 1]

11 Mar 2020

please see the letter; I have also revised the supplemental materials accordingly

---

## [Editor Report · Decision Letter 2]

1 Apr 2020

Distinguishing moral hazard from access for high-cost healthcare under insurance

PONE-D-19-27471R2

Dear Dr. Robertson,

We are pleased to inform you that your manuscript has been judged scientifically suitable for publication and will be formally accepted for publication once it complies with all outstanding technical requirements.

With kind regards,

Jason Anthony Aimone

Academic Editor

PLOS ONE
---

## [Editor Report · Acceptance letter]

3 Apr 2020

PONE-D-19-27471R2 

Distinguishing moral hazard from access for high-cost healthcare under insurance 

Dear Dr. Robertson:

I am pleased to inform you that your manuscript has been deemed suitable for publication in PLOS ONE. Congratulations! Your manuscript is now with our production department. 

With kind regards,

on behalf of

Dr. Jason Anthony Aimone 

Academic Editor

PLOS ONE